# Serum Asprosin Correlates with Indirect Insulin Resistance Indices

**DOI:** 10.3390/biomedicines11061568

**Published:** 2023-05-28

**Authors:** Małgorzata Mirr, Anna Braszak-Cymerman, Aleksandra Ludziejewska, Matylda Kręgielska-Narożna, Paweł Bogdański, Wiesław Bryl, Maciej Owecki

**Affiliations:** 1Department of Public Health, Poznan University of Medical Sciences, Rokietnicka St. 4, 60-806 Poznań, Poland; mowecki@ump.edu.pl; 2The Doctoral School of the Poznan University of Medical Sciences, Bukowska St. 70, 60-812 Poznań, Poland; 3Department of Internal Diseases, Metabolic Disorders and Hypertension, Poznan University of Medical Sciences, Szamarzewskiego St. 84, 60-569 Poznań, Poland; anna.braszak@gmail.com (A.B.-C.); wieslawbryl@wp.pl (W.B.); 4Department of Laboratory Diagnostics, Poznan University of Medical Sciences, Szamarzewskiego St. 84, 60-569 Poznań, Poland; aludziejewska@ump.edu.pl; 5Department of Treatment of Obesity, Metabolic Disorders and Clinical Dietetics, Poznan University of Medical Sciences, Szamarzewskiego St. 84, 60-569 Poznań, Poland

**Keywords:** asprosin, adipokine, obesity, insulin resistance

## Abstract

Background and Objectives: Insulin resistance is a major contributor to the development of type 2 diabetes and can be assessed using indirect indicators calculated from non-invasive tests. Asprosin is a recently discovered adipokine with a postulated effect on glycemic regulation. This study aimed to investigate the correlation between serum asprosin levels and insulin resistance indices. The correlation between circulating asprosin and obesity indices was also investigated. Materials and Methods: A total of 50 non-diabetic patients with obesity and 50 healthy volunteers were studied. Laboratory data, including circulating asprosin and anthropometric data, were collected. The following insulin resistance indices were calculated: triglyceride–glucose index (TyG), TyG–neck circumference (TyG–NC), TyG–neck circumference to height ratio (TyG–NHtR), TyG–waist circumference (TyG–WC), TyG–waist to height ratio (TyG–WHtR), TyG–body mass index (TyG–BMI), and the ratio between triglycerides and high-density cholesterol (TG/HDLc). The obtained data were analyzed separately for males and females. Results: Asprosin concentrations were significantly higher in obese patients (*p* < 0.001). Asprosin concentrations positively correlated with body mass index (*p* < 0.001, r = 0.8 in females and r = 0.8 in males), waist circumference (*p* < 0.001, r = 0.73 in females and r = 0.81 in males), and all tested indices of insulin resistance. The strongest correlation was observed for TyG–BMI (*p* < 0.001, r = 0.78 in females and r = 0.81 in males). Circulating asprosin was higher in females (*p* < 0.001). Conclusions: Asprosin can be considered a marker of obesity and insulin resistance.

## 1. Introduction

The prevalence of obesity and diabetes is increasing rapidly in developed countries [1]. Insulin resistance, an axial disorder of the metabolic syndrome and a crucial risk factor for type 2 diabetes, can be assessed using several methods [2]. Several indirect indices of insulin resistance based on biochemical and anthropometric data have been proposed [2]. The triglyceride–glucose index (TyG) calculated from fasting glucose and triglyceride concentrations was presented as a valuable tool in identifying patients with insulin resistance with high specificity and sensitivity [3]. TyG strongly correlates with insulin resistance assessed by the hyperinsulinemic–euglycemic clamp [3]. Several indices derived from TyG and anthropometric indices, including TyG–waist circumference (TyG–WC), TyG–waist to height ratio (TyG–WHtR), and TyG–BMI with diagnostic accuracy in insulin resistance have been proposed [4]. In a previous study, we proposed two new markers of insulin resistance, TyG–neck circumference (TyG–NC) and TyG–neck circumference to height ratio (TyG–NHtR), and demonstrated their usefulness in diagnosing metabolic syndrome, the main feature of which is insulin resistance [5]. A simple index defined as the ratio between triglycerides and high-density cholesterol (TG/HDLc) is also associated with metabolic syndrome and risk of type 2 diabetes [6,7]. Some adipokines are emerging as promising markers of reduced insulin sensitivity [8].

Asprosin, a novel adipokine identified by Romere et al. in 2016, is the C-terminal cleavage product of profibrillin encoded by the FBN1 gene [9]. Asprosin is secreted by white adipose tissue in response to low glycemia [9]. It has been reported that asprosin activates protein kinase A (PKA) in the liver, followed by the release of glucose from hepatocytes [9]. Insulin reverses this effect by reducing PKA activity via the cyclic AMP system [9]. Hyperlipidemia stimulates asprosin mRNA expression in adipocytes [10]. Previous reports suggest that asprosin may be an adipokine contributing to the increase in insulin resistance by enhancing inflammation as asprosin treatment causes elevated endoplasmic reticulum inflammation markers in mouse skeletal muscle cells [10]. Asprosin causes impairment of insulin signaling in differentiated skeletal muscle cells by reducing insulin receptor substrate 1 and Akt phosphorylation [10]. Li et al. identified a potential receptor for asprosin, olfactory receptor OLFR734 [11]. Olfr734 knockout mice showed a blunted response to asprosin, cAMP levels, and hepatic glucose production [11]. An enhanced insulin responsiveness in hyperinsulinemic–euglycemic clamp tests was observed in Olfr734 knockout mice [11]. Moreover, a single subcutaneous injection of recombinant asprosin results in an immediate spike in glucose level and compensatory hyperinsulinemia [9]. Asprosin participates in appetite regulation by directly stimulating orexigenic neurons in the hypothalamus [12]. Asprosin level is elevated in obesity, and it increases in accordance with increasing body mass index (BMI) [13,14,15,16,17]. Asprosin concentration correlates positively with HOMA-IR (homeostatic model assessment for insulin resistance), an index indirectly assessing insulin resistance [14,16,17,18,19,20,21]. However, another study in the pediatric population did not confirm this relationship, and one study reported a negative correlation between asprosin and HOMA-IR [22,23]. Higher asprosin levels are found in diabetes mellitus type 2 [17,18,19,20,24,25]. Circulating asprosin levels correlate positively with fasting glucose, insulin, and glycated hemoglobin [14,16,17,19,20,24,25].

The dependence of asprosin concentrations on female sex hormones has also been suggested, and higher concentrations of asprosin have been reported in polycystic ovary syndrome independently of body mass index [26]. This relationship may result from increased peripheral insulin resistance associated with this syndrome [25,27].

Previous reports suggest that elevated levels of asprosin may be associated with reduced insulin sensitivity. This study aimed to check the correlation between asprosin serum concentration and non-invasive indices of insulin resistance, such as TyG, TyG–NC, TyG–NHtR, TyG–WC, TyG–WHtR, TyG–BMI, and TG/HDLc in patients with and without obesity. The correlation between circulating asprosin and obesity indices was also investigated. Analyses were performed separately for males and females.

## 2. Materials and Methods

### 2.1. Study Group

A total of 100 adult patients from the Department of Internal Medicine, Metabolic Diseases & Hypertension, Poznan University of Medical Sciences were included in the research. This included 50 non-diabetic patients with diagnosed obesity (defined as body mass index exceeding 30 kg/m^2^) and 50 participants without obesity that were recruited as controls. Secondary causes of obesity and the use of drugs affecting carbohydrate metabolism (glucocorticoids, neuroleptics, thiazides, medroxyprogesterone acetate, beta-blockers) were excluded in all patients. The exclusion criteria included acute or chronic inflammatory disease, history of chronic disease of the liver, heart, kidneys, or endocrine glands (except for insulin resistance), history of neoplasms, and current pregnancy.

### 2.2. Anthropometric Measurements

Body mass and height were measured while participants were in underwear and fasting, and the body mass index (BMI) was calculated as the body mass divided by the square of the body height (in units of kg/m^2^). Neck circumference (NC) was measured using a plastic tape from the level just below the laryngeal prominence perpendicular to the long axis of the neck with the head parallel to the Frankfort plane. Waist circumference (WC) was measured at the approximate midpoint between the lower margin of the last palpable rib and the top of the iliac crest, using plastic tape.

Anthropometric indices were calculated as follows:-waist to height ratio (WHtR) was defined as the waist circumference divided by the body height,-neck to height ratio (NHtR) was defined as the neck circumference divided by the body height.

### 2.3. Laboratory Tests and Insulin Resistance Indices Calculation

Each patient was tested for concentrations of serum glucose, total cholesterol (TC), HDL cholesterol (HDLc), LDL cholesterol (LDLc), and triglycerides (TG). Glycated hemoglobin levels were available for 29 participants.

Asprosin serum concentrations were measured using a commercially available ELISA kit (EH4176, Wuhan Fine Biotech Co., Ltd., Wuhan, China, intra-assay coefficient of variation below 8%) according to the manufacturer’s instructions.

Insulin resistance indices were calculated as follows:

TyG  =  Ln [fasting TG (mg/dL)  ×  FPG (mg/dL)/2] [28],TG/HDLc  =  fasting TG (mmol/L)/fasting HDL cholesterol (mmol/L) [7],TyG–NC  =  TyG  ×  NC [5],TyG–NHtR  =  TyG  ×  NHtR [5],TyG–WC = TyG × WC [29],TyG–WHtR = TyG × WHtR [30],TyG–BMI = TyG × BMI [29].

### 2.4. Statistics

The statistical analysis was carried out using PQ Stat v. 1.8.2 software. The normality of distribution was tested with the Shapiro–Wilk test. Among the analyzed variables, only TyG, TyG–WC, and TyG–WHtR showed a normal distribution in both groups. Mann–Whitney U test, Student’s *t*-test, and Spearman’s rank coefficient were performed. The usefulness of asprosin in distinguishing obese patients from non-obese patients was determined using the ROC curve. The area under the ROC (AUC) was calculated. The optimal cut-off point was selected according to Youden’s index. A significance level of 0.05 was assumed. The original data set is available in the Appendix A.

### 2.5. Ethical Approval

This research was approved by the Poznań University of Medical Sciences Bioethical Committee (number of approval 1152/19).

## 3. Results

Table 1 presents the basic clinical data of the study group. Females constituted 60% of participants (in the group with obesity, there were 62% females, and in the group without obesity, 58%). There were significant differences between the obese and non-obese group, as presented in Table 1. Asprosin concentration in the group with obesity was significantly higher than in the control group.

Table 2 summarizes the clinical data of females and males.

Figure 1 shows the levels of serum asprosin divided by sex and the presence of obesity.

Spearman’s rank correlations between circulating asprosin and other variables are presented in Table 3. We observed a positive correlation between circulating asprosin and glycated hemoglobin (r = 0.45, *p* = 0.015). Spearman’s correlations for asprosin and other variables for females and males are presented in Table 4 and Table 5, respectively.

The area under the ROC curve of asprosin for predicting the presence of obesity, 95% confidence intervals, optimal cut-off points, corresponding sensitivities, and specificities for the whole group and both sexes are shown in Table 6. Figure 2 presents the receiver operating curves of asprosin concentrations for detecting obesity.

## 4. Discussion

Our results confirmed previous reports of higher asprosin concentrations in obese patients [16]. Similar to previous studies, asprosin positively correlated with body mass and BMI [13,16,21,31]. We observed a positive correlation between circulating asprosin and BMI in both obese and non-obese patients, with the association being more substantial in the obese group. The ROC curve analysis showed the usefulness of asprosin in distinguishing obese patients from healthy ones with relatively high sensitivity and specificity. As Romere et al. showed, asprosin is secreted by white adipose tissue cells, which explains the above relationships [9]. The correlation between asprosin and BMI was stronger in each of the sexes analyzed separately, with a particularly significant relationship for men with obesity (r = 0.9). Consistent with previous findings, asprosin concentration positively correlated with waist circumference in the whole group [20,21,32]. The relationship between asprosin and WC was especially strong in the group of men. Among the anthropometric indicators, BMI and WHtR showed the most substantial relationship with asprosin in the whole group. However, in the group of women we observed the strongest relationship for body mass and BMI, and in the group of men for body mass and waist circumference. These findings underline the need to analyze anthropometric data for each sex separately, which results from the different distribution of body fat in men and women, as at comparable levels of total adiposity, females have more subcutaneous adipose tissue than males [33,34,35].

Despite similar BMI, we observed significantly higher asprosin concentrations in women than in men, which may be due to the higher percentage and different distribution of white adipose tissue in women [36,37]. The effects of the menstrual cycle and oral contraceptive use on asprosin concentrations have been shown [27]. However, in our study, most of the women were postmenopausal, and the effect of female sex hormones may not have been significant.

Asprosin positively correlated with all TyG-derived insulin resistance indices and TG/HDLc, with the strongest correlation observed for TyG–BMI and TyG–WHtR. In the group of women, the strongest correlation was achieved by TyG–BMI, and in the group of men by TyG–BMI and TyG–WC. We hypothesize that these results may be due to a greater effect of increased body mass, and consequently, anthropometric indices, on asprosin concentrations. Interestingly, we presented a stronger correlation for TyG–NHtR and TyG–NC than for TyG alone in both sexes, highlighting the importance of neck circumference as a significant indicator of obesity. However, the correlation of asprosin with TyG and TG/HDLc, which do not take into account anthropometric indices, was also present, suggesting that asprosin may be considered one of the markers of insulin resistance regardless of body mass.

Notably, we obtained significant correlations between asprosin and TyG–BMI, TyG–WC, and TyG–NC for females and males without obesity. It seems that serum asprosin correlates with indirect indices of insulin resistance independently of BMI, which makes asprosin a potential early marker of insulin resistance. The importance of asprosin as an early marker of insulin resistance requires further research with direct measurement of insulin sensitivity. We should also mention normal weight individuals with insulin resistance, the so-called metabolically obese normal weight individuals, which exhibit various degrees of glucose tolerance [3]. In the current study, the non-obese group could also include patients with insulin resistance, which may explain the significant correlation between circulating asprosin and insulin resistance indices in this group.

A study by Zhong et al. presented a correlation between asprosin and TyG with r = 0.327; however, the research was conducted with diabetic patients [38]. Therefore, it seems that the correlation between TyG and asprosin is present in both non-diabetic and diabetic patients. A study by Zhang et al. showed that the concentration of asprosin was significantly higher in patients with type 2 diabetes compared to healthy BMI-matched patients [24]. However, in this study, waist circumference was significantly higher in diabetic patients despite similar BMI, indicating a different distribution of body fat in these groups [24].

We observed a positive correlation of asprosin with glycated hemoglobin, which expresses the average glucose concentration for the last approximately 120 days, which may confirm the involvement of asprosin in inducing glucose release [9]. Asprosin levels increase in fasting condition and decrease in a postprandial state [9]. Corica et al. studied meal-related changes in asprosin levels in obese children without diabetes [39]. There was no significant difference between fasting and 2 h postprandial asprosin serum levels in the whole study group [39]. The authors reported that more than half of the group was marked by an unexpected increase in 2 h postprandial asprosin, and the remaining participants showed a decrease in asprosin levels after a meal [39]. Patients with a significant increase in postprandial asprosin had higher fasting blood glucose levels and more frequently impaired fasting glucose compared to patients who had a significant decrease in postprandial asprosin levels [39]. The authors suggested that altered asprosin secretion may be an early marker of insulin resistance [39]. Further research is required to confirm these findings.

Several studies have shown a positive correlation between asprosin and HOMA-IR in both diabetic and non-diabetic patients [17,18,19,20,21]. A positive correlation between asprosin and HOMA-IR has also been reported in children and adolescents [14,16]. However, several studies have investigated the association of asprosin with obesity in children, and the results are conflicting [40]. In studies by Sünnetçi Silistre et al. and Liu et al., higher concentrations of asprosin in obese children were presented [41,42]. In the study by Liu et al., circulating asprosin was significantly elevated in obese children with non-alcoholic fatty liver disease (NAFLD) when compared to obese children without NAFLD [42]. However, there was no difference in asprosin concentrations between obese children without NAFLD and lean children [42]. Moreover, two studies showed lower levels of asprosin in obese children [22,23]. Long suggested that the degree of obesity may be a major factor causing these discrepancies [23]. He postulated that in children with a lower degree of obesity, metabolic disorders are compensated with a reduced concentration of asprosin [23]. When compensatory mechanisms are no longer sufficient, there is a further increase in body mass along with increase in asprosin concentration, which additionally intensifies insulin resistance [23]. It is possible that the relationship between asprosin and body mass undergoes a dynamic process depending on the severity of obesity and, consequently, metabolic disorders [23,42]. However, clarification of these inconsistencies requires further research.

To the best of the authors’ knowledge, this is the first study presenting the correlation of asprosin with indirect insulin resistance indices in a group of non-diabetic patients. However, some limitations of this study should be noted. First, this study was conducted on relatively few samples from only one center, so it cannot be extrapolated to other populations. Estimated cut-off points due to the small group should be interpreted cautiously and validated in a larger population. Finally, this study was observational and does not prove causality.

## Figures and Tables

**Figure 1 biomedicines-11-01568-f001:**
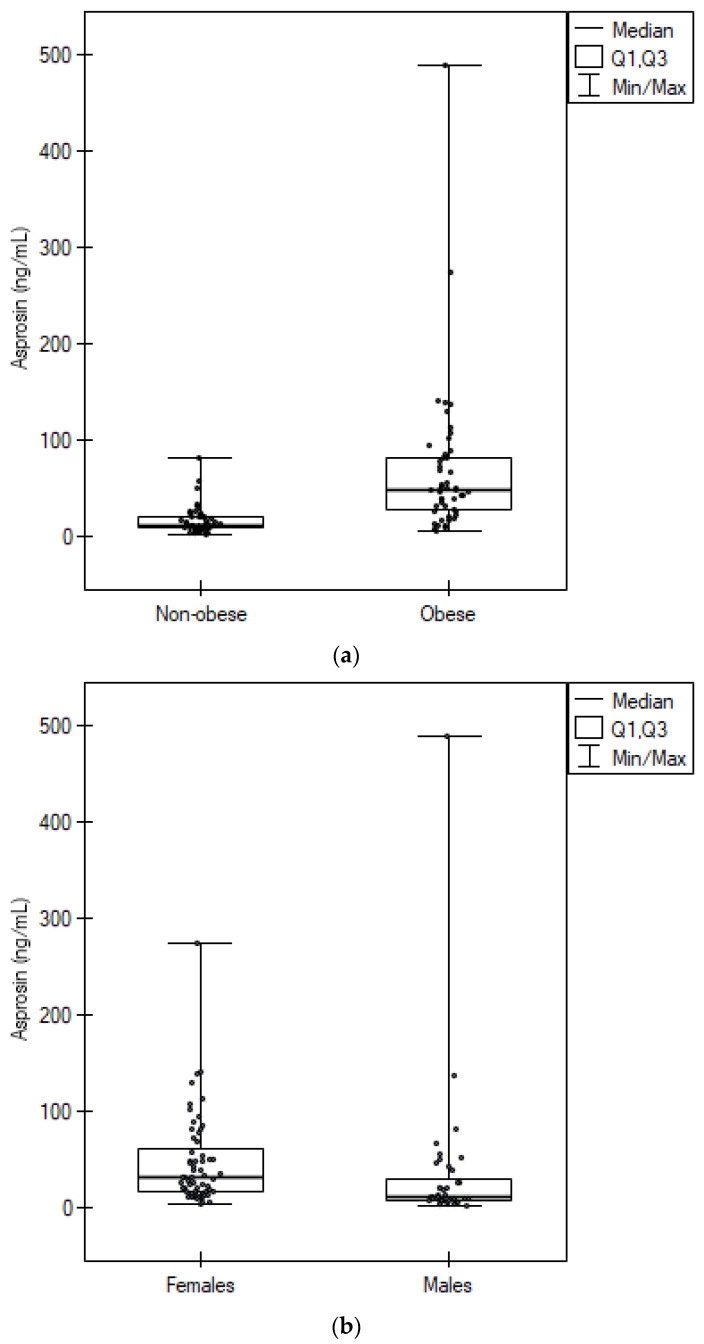
Serum asprosin levels. (**a**) comparison of obese and non-obese, (**b**) comparison of females and males.

**Figure 2 biomedicines-11-01568-f002:**
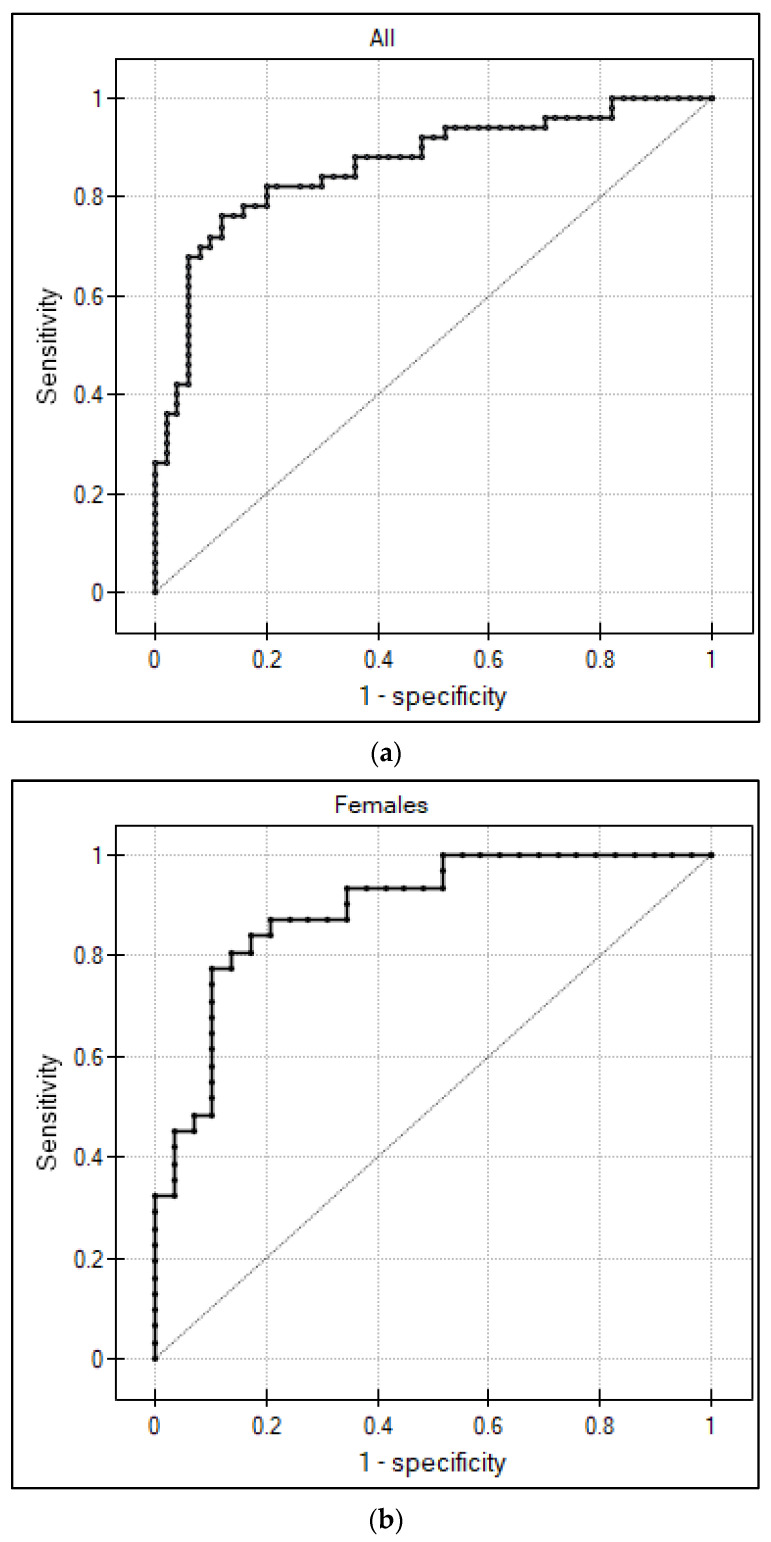
Receiver operating curve of serum asprosin for predicting the presence of obesity. (**a**) all participants, (**b**) females, (**c**) males.

**Table 1 biomedicines-11-01568-t001:** Group characteristics. Data are reported as medians (interquartile range) except for TyG, TyG-WC, and TyG–WHtR, which are presented as mean ± SD.

Variable	Study Group (N = 50)	Controls (N = 50)	*p*
Age (years)	57.5 (45.0–65.0)	62.0 (43.0–66.0)	0.548
Body mass (kg)	98.5 (84.3–115.7)	73.9 (62.3–82.4)	<0.001
BMI (kg/m^2^)	34.4 (31.6–41.3)	26.0 (23.6–28.3)	<0.001
NC (cm)	41.0 (37.0–44.0)	37.0 (33.3–39.0)	<0.001
WC (cm)	110.5 (100.1–128.3)	91.0 (83.3–100.0)	<0.001
NHtR	0.24 (0.23–0.26)	0.22 (0.20–0.23)	<0.001
WHtR	0.67 (0.63–0.76)	0.55 (0.49–0.59)	<0.001
TyG	9.0 ± 0.6	8.6 ± 0.7	0.003
TyG–NC	368.1 (330.0–413.4)	307.7 (285.6–355.4)	<0.001
TyG–NHtR	2.2 (2.0–2.4)	1.9 (1.7–2.1)	<0.001
TyG–WC	1052.0 ± 194.7	787.3 ± 129.3	<0.001
TyG–WHtR	6.3 ± 1.0	4.7 ± 0.7	<0.001
TyG–BMI	309.6 (290.4–370.9)	225.5 (193.6–244.4)	<0.001
TG/HDLc	1.4 (0.9–2.4)	0.8 (0.4–1.7)	0.002
Asprosin (ng/mL)	47.8 (27.3–81.6)	11.3 (8.5–20.4)	<0.001

**Table 2 biomedicines-11-01568-t002:** Group characteristics for females and males. Data are reported as medians (interquartile range) except for TyG, TyG–WC, and TyG–WHtR, which are presented as mean ± SD. NS—not significant.

Variable	Females (N = 60)	Males (N = 40)	*p*
Age (years)	60.0 (50.5–67.0)	53.5 (39.75–64.0)	0.034
Body mass (kg)	75.2 (65.6–87.6)	92.5 (82.9–119.2)	<0.001
BMI (kg/m^2^)	30.1 (25.2–32.8)	29.6 (27.3–37.7)	NS
NC (cm)	35.3 (33.4–39.0)	42.0 (39.0–45.0)	<0.001
WC (cm)	95.5 (85.0–106.3)	107.5 (100.8–122.3)	<0.001
NHtR	0.22 (0.20–0.24)	0.24 (0.22–0.26)	<0.001
WHtR	0.59 (0.54–0.65)	0.60 (0.58–0.71)	NS
TyG	8.8 ± 0.7	8.9 ± 0.7	NS
TyG–NC	312.1 (285.6–347.0)	375.2 (349.8–416.6)	<0.001
TyG–NHtR	1.9 (1.8–2.1)	2.1 (2.0–2.3)	<0.001
TyG–WC	850.9 ± 172.2	1022.9 ± 224.7	<0.001
TyG–WHtR	5.3 ± 1.0	5.8 ± 1.2	0.021
TyG–BMI	258.5 (211.8–299.4)	276.0 (238.1–361.7)	NS
TG/HDLc	1.0 (0.5–1.7)	1.5 (0.7–2.7)	0.043
Asprosin (ng/mL)	30.9 (16.2–60.7)	10.8 (7.8–29.7)	<0.001

**Table 3 biomedicines-11-01568-t003:** Spearman’s rank coefficient results for serum asprosin and other variables. NS—not significant.

	All (N = 100)		Study Group (N = 50)		Controls (N = 50)	
Variable	*r*	*p*	*r*	*p*	*r*	*p*
Body mass (kg)	0.47	<0.001	0.30	0.034	0.01	NS
BMI (kg/m^2^)	0.69	<0.001	0.48	<0.001	0.28	0.049
NC(cm)	0.15	NS	0.01	NS	−0.26	NS
WC (cm)	0.47	<0.001	0.26	NS	0.06	NS
NHtR	0.28	0.005	0.10	NS	−0.12	NS
WHtR	0.60	<0.001	0.36	0.011	0.22	NS
TyG	0.25	0.014	−0.08	NS	0.23	NS
TyG–NC	0.25	0.014	−0.02	NS	−0.07	NS
TyG–NHtR	0.32	0.001	0.00	NS	0.00	NS
TyG–WC	0.48	<0.001	0.20	NS	0.15	NS
TyG–WHtR	0.58	<0.001	0.28	NS	0.24	NS
TyG–BMI	0.64	<0.001	0.35	0.013	0.28	0.047
TG/HDLc	0.24	0.018	0.00	NS	0.11	NS

**Table 4 biomedicines-11-01568-t004:** Spearman’s rank coefficient results for serum asprosin and other variables in females. NS—not significant.

	All (N = 60)		Obesity (N = 31)		Non-obese (N = 29)	
Variable	*r*	*p*	*r*	*p*	*r*	*p*
Body mass (kg)	0.81	<0.001	0.56	0.001	0.70	<0.001
BMI (kg/m^2^)	0.80	<0.001	0.51	0.004	0.69	<0.001
NC(cm)	0.54	<0.001	0.22	NS	0.37	0.049
WC (cm)	0.73	<0.001	0.40	0.028	0.70	<0.001
NHtR	0.45	<0.001	0.09	NS	0.27	NS
WHtR	0.69	<0.001	0.28	NS	0.58	<0.001
TyG	0.40	0.002	0.01	NS	0.34	NS
TyG–NC	0.63	<0.001	0.20	NS	0.59	<0.001
TyG–NHtR	0.57	<0.001	0.03	NS	0.46	0.011
TyG–WC	0.75	<0.001	0.37	0.043	0.73	<0.001
TyG–WHtR	0.70	<0.001	0.22	NS	0.62	<0.001
TyG–BMI	0.78	<0.001	0.44	0.014	0.71	<0.001
TG/HDLc	0.43	<0.001	0.12	NS	0.34	NS

**Table 5 biomedicines-11-01568-t005:** Spearman’s rank coefficient results for serum asprosin and other variables in males. NS—not significant.

	All (N = 40)		Obesity (N = 19)		Controls (N = 21)	
Variable	*r*	*p*	*r*	*p*	*r*	*p*
Body mass (kg)	0.84	<0.001	0.85	<0.001	0.69	<0.001
BMI (kg/m^2^)	0.80	<0.001	0.90	<0.001	0.50	0.020
NC(cm)	0.64	<0.001	0.68	0.001	0.23	NS
WC (cm)	0.81	<0.001	0.84	<0.001	0.58	0.006
NHtR	0.51	<0.001	0.66	0.002	−0.07	NS
WHtR	0.75	<0.001	0.88	<0.001	0.31	NS
TyG	0.25	NS	−0.29	NS	0.54	0.012
TyG–NC	0.63	<0.001	0.49	0.034	0.58	0.006
TyG–NHtR	0.50	<0.001	0.33	NS	0.28	NS
TyG–WC	0.81	<0.001	0.82	<0.001	0.64	0.002
TyG–WHtR	0.77	<0.001	0.83	<0.001	0.42	NS
TyG–BMI	0.81	<0.001	0.87	<0.001	0.61	0.004
TG/HDLc	0.28	NS	−0.10	NS	0.50	0.021

**Table 6 biomedicines-11-01568-t006:** ROC analysis of serum asprosin for predicting the presence of obesity. AUC—area under the curve.

	AUC	Cut-Off	Sensitivity	Specificity	*p*
All (N = 100)	0.864	26.77	0.76	0.88	<0.001
Females (N = 60)	0.891	34.55	0.77	0.90	<0.001
Males (N = 40)	0.864	11.19	0.84	0.81	<0.001

## Data Availability

The original data set is available in the Appendix A.

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
