# Peer review of "Serum Asprosin Correlates with Indirect Insulin Resistance Indices"

_biomedicines, 2023, doi:10.3390/biomedicines11061568_

Round 1
Reviewer 1 Report
In the present study, Mirr and colleagues assessed whether serum asprosin levels correlate with different insulin resistance indexes. To this end, the authors analyzed serum asprosin levels and different parameters related with insulin resistance in 50 (29 females and 21 males) non-obese healthy individuals and 50 (31 females and 19 males) obese non-diabetic subjects. According to their analysis the authors suggest that serum asprosin levels positively correlate with BMI, waist circumference, TyG-BMI, and all tested indirect insulin resistance indexes. Based in this analysis the authors suggest that asprosin can be consider a marker of obesity and insulin resistance. This is a short study, mainly descriptive in nature, and confirmatory of previous work. To support their suggestions, several issues need to be clarified further.
A major limitation of this work is the fact that serum asprosin levels correlate positively with several insulin resistance indexes in non-obese healthy and obese non-diabetic subjects, both in females and males. According to data in Tables 4 and 5, a positive correlation between serum asprosin levels and BMI, WC, TyG-WC, TyG-BMI, is found in non-obese healthy and obese non-diabetic subjects, both in females and males. Hence, how can be explained that serum asprosin levels could be consider a marker of insulin resistance when a positive correlation is found in both, obese insulin resistant and in non-obese healthy subjects? These discrepancies should be discussed.
On the other hand, HOMA-IR index it is generally used as an index for insulin resistance. In fact, the authors stated that asprosin concentration correlates positively with HOMA-IR. Hence, which was the aim to assess the correlation between asprosin levels and other indirect insulin resistance indexes? This aspect should be thoroughly discussed in the manuscript. In this sense, in p. 9, lines 235-236, the authors conclude that the correlation between asprosin and TyG may be stronger in patients with higher insulin resistance. Nevertheless, to support this suggestion further work is necessary.
Reviewer 2 Report
In the discussion (line 198,199) the Authors cite literature reference 16 and 14 (line 244) as confirming reference points for their results.Unfortunately, the results of this publication refer to a population of children and adolescents. In the studies performed in this population, the results did not demonstrate a homogeneous and comparable response of asprosin compared with adults .In fact, different results of plasma asprosin concentration are described both in baseline condition and after metabolic stimuli.
It is recommended that the Authors expand the literature review and examine the results more critically in the pediatric field as well, discussing separately the results in the pediatric setting found in the literature (doi: 10.3389/fendo.2022.1101091, https://doi.org/10.1016/j.cyto.2021.155477, https://doi.org/10.1159/000500523, , https://doi.org/10.1111/ped.14176, https://doi.org/10.3389/fendo.2021.805700) considering the pediatric population, which is often ideal , as it is free from chronic complications .
These considerations are aimed at enhancing the interesting results of this manuscript.
Round 2
Reviewer 1 Report
The revised version of the manuscript has been improved and most of reviewer's comments were addressed with adequate explanations. As minor comment, I would suggest including a statement and references, either in the Introduction section and/ or in the Discussion section, regarding the presence of decreased insulin sensitivity and/or increased insulin resistance in non-obese individuals.
